# RESEARCH REPORT

# Bulk-level maps of pioneer factor binding dynamics during the *Drosophila* maternal-to-zygotic transition

Sadia Siddika Dima[1] and Gregory T. Reeves[1,2,*]

## ABSTRACT

Gene regulation by transcription factors (TFs) binding cognate sequences is of paramount importance. For example, the TFs Zelda (Zld) and GAGA factor (GAF) are widely acknowledged for pioneering gene activation during zygotic genome activation (ZGA) in *Drosophila*. However, quantitative dose/response relationships between bulk TF concentration and DNA binding, an event tied to transcriptional activity, remain elusive. Here, we map these relationships during ZGA: a crucial step in metazoan development. To map the dose/response relationship between nuclear concentration and DNA binding, we performed raster image correlation spectroscopy, a method that can measure biophysical parameters of fluorescent molecules. We found that, although Zld concentration increases during nuclear cycles 10 to 14, its binding in the transcriptionally active regions decreases, consistent with its function as an activator for early genes. In contrast, GAF-DNA binding is nearly linear with its concentration, which sharply increases during the major wave, implicating its involvement in the major wave. This study provides key insights into the properties of the two factors and puts forward a quantitative approach that can be used for other TFs to study transcriptional regulation.

KEY WORDS: Pioneer factors, Zygotic genome activation, Zelda, GAGA factor, DNA binding, Raster image correlation spectroscopy

## INTRODUCTION

In *Drosophila*, zygotic genome activation (ZGA) begins with the transcription of a handful of genes during its minor wave, followed by a major wave when thousands of genes are transcribed (Harrison et al., 2023; Pritchard and Schubiger, 1996; Schulz and Harrison, 2019; Vastenhouw et al., 2019). The transcription factor (TF) Zelda (Zld) has the ability to bind nucleosomal DNA (Garcia et al., 2019; McDaniel et al., 2019) and subsequently to facilitate the binding of other TFs (Satija and Bradley, 2012; Li et al., 2014; Foo et al., 2014; Sun et al., 2015; Schulz et al., 2015; Yáñez-Cuna et al., 2012; Xu et al., 2014), which are the two defining features of a special class of TFs known as pioneer factors (Iwafuchi-Doi and Zaret, 2014; Zaret and Carroll, 2011; Zaret and Mango, 2016). The maternally

encoded TF GAGA factor (GAF) also possesses pioneer-like properties (Blythe and Wieschaus, 2015, 2016; Gaskill et al., 2021; Judd et al., 2021; Schulz et al., 2015; Sun et al., 2015).

The relationship between the concentration of TFs such as these pioneer factors and gene expression remains an open question (Ay and Arnosti, 2011; Kim and O'Shea, 2008). Fluorescent imaging of live and fixed tissues can give relative concentrations of TFs, which are often used to predict TF activity by assuming some nonlinear relationship between concentration and activity (Jaeger et al., 2004; Kanodia et al., 2012; Manu et al., 2009; O'Connell and Reeves, 2015; Papatsenko and Levine, 2005, 2011; Zinzen et al., 2006). However, validation of these relationships using dynamic quantitative data in live cells is currently lacking. Furthermore, the binding of the TFs to DNA – which is required for transcriptional regulation – is dependent not only on their nuclear concentration, but also on factors such as chromatin accessibility and saturation kinetics. On the other hand, recent studies have identified hubs of TF binding at DNA sites, which may provide more direct measures of TF activity (Dufourt et al., 2018; Mir et al., 2017, 2018; Yamada et al., 2019). Thus, knowing only total concentration is inadequate to predict TF binding, and direct measurements of TF hubs lack generalizability; an input/output map between the two, based on quantitative measurements of concentration and binding, is required.

To bridge this gap, we used raster image correlation spectroscopy (RICS) to quantify the concentration and binding of the pioneer-like factors Zld and GAF in live embryos over the course of nuclear cycles (ncs) 10-14 (Al Asafen et al., 2024; Brown et al., 2008; Digman and Gratton, 2009; Digman et al., 2005a,b; Schloop et al., 2025). These measurements allowed us to construct dose/response relationships suggesting that the binding of Zld to transcriptionally active sites is dependent on factors other than its concentrations alone, whereas the GAF concentration is the primary driver of its binding. Furthermore, we found that GAF must bind and saturate its sites in the inactive regions of the DNA before it can bind to active DNA sites, resulting in a delay in its role in ZGA. Similar approaches can be used to obtain a comprehensive quantitative picture of dynamics of other TFs for gene regulation studies.

## RESULTS AND DISCUSSION

Raster-scanned confocal images have a fast-scanning direction (pixel-to-pixel) and a slow-scanning direction (line retracing; Fig. 1A). RICS analysis uses intensity fluctuations of GFP-tagged molecules, correlated in time and space, to build an autocorrelation function (ACF) in two dimensions, corresponding to the fast ($\Delta x$) and slow ($\Delta y$) scanning directions, respectively (Fig. 1B; see Materials and Methods). The amplitude of the ACF, $A$, is inversely proportional to the GFP concentration (Fig. 1C), while the shape of the ACF, especially in the slow direction, is determined by the fraction of GFP that is freely diffusible versus that which is immobile (Fig. 1D). Furthermore, cross-correlations in the fluctuations between GFP and RFP (tagged to His2Av) allow us to calculate the fraction of

[1]Department of Chemical Engineering, Texas A&M University, College Station, TX 77843, USA. [2]Faculty in Genetics and Genomics, Texas A&M University, College Station, TX 77843, USA.

*Author for correspondence (gtreeves@tamu.edu)

S.S.D., 0009-0008-3558-9093; G.T.R., 0000-0003-0836-7766

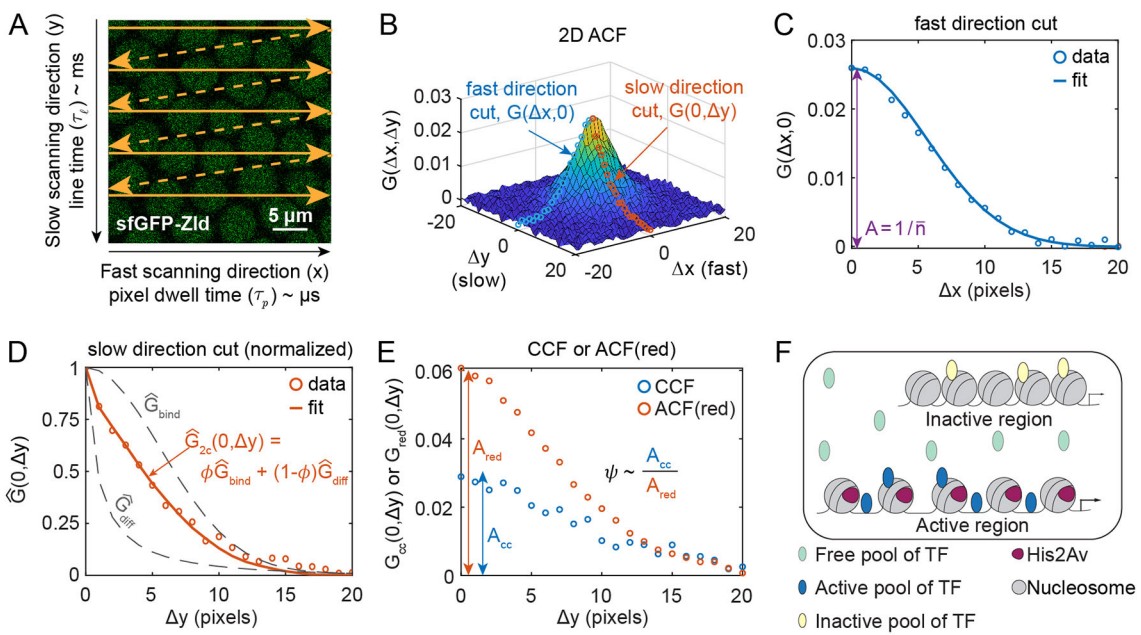

**Fig. 1. Raster Image Correlation Spectroscopy (RICS).** (A) Laser scanning confocal microscopes build images by raster scan, with a fast scanning direction (*x* direction, solid arrows) and a slow scanning direction (*y* direction) due to line retracing (dashed arrows). Image shows a mid-nc14 embryo expressing sfGFP-Zld. (B) Two-dimensional autocorrelation function (ACF) from the embryo depicted in A. Cuts along the fast (blue circles) and slow (orange circles) directions are depicted. (C) Cut of ACF along the fast direction. Solid curve: fit of Gaussian-shaped PSF, used to estimate the ACF amplitude, *A*. (D) Plot of the slow direction data (circles) and the fit to the slow direction (solid curve), composed of a linear combination between two ACFs (gray dotted curves): an immobile ACF ($\widehat{G}_{bind}$) and a diffusible ACF ($\widehat{G}_{diff}$). The linear combination weight is $\phi$, the immobile fraction. (E) Plots of cuts of the cross correlation function (CCF; blue circles) and of the ACF in the red channel (orange circles). The ratio of the amplitudes ($A_{cc}/A_{red}$) is proportional to $\psi$, the fraction bound in active regions of the DNA. (F) Illustration of the different pools of TF: freely diffusible (light blue), bound to active regions (dark blue) and bound to inactive regions (yellow). His2Av (purple) is associated with the active regions of DNA.

GFP-tagged molecules that are bound to the same structure as the His2Av-RFP (likely to be DNA; Bacia and Schwille, 2007; Weidemann et al., 2002; Fig. 1E). Therefore, RICS allows us to quantify not only the absolute concentration of GFP-tagged molecules, but also the fraction that is freely diffusible, the immobile fraction, and the fraction correlated to His2Av (Fig. 1F).

## Zld levels bound to DNA decrease while nuclear concentration increases

We performed RICS analysis on the nuclei of blastoderm-stage embryos expressing sfGFP-Zld (Hamm et al., 2017) (Fig. 2A and Movie 1) to measure the dynamics of the nuclear concentration of Zld over time. We found the total nuclear concentration of Zld increased from one nuclear cycle to the next during nc 10 to 14 (Fig. 2B). Increase in Zld levels has been reported previously using immunoblotting (Harrison et al., 2010; Nien et al., 2011). Furthermore, the nuclear concentrations of Zld vary significantly within each nc, as more sfGFP-Zld enters nuclei after mitosis and fills it during the interphase. The longer duration of nc 14 allows Zld levels to reach a steady state, unlike the other, shorter, ncs. In contrast, we observed that the nuclear Zld concentration drastically decreases from the end of one interphase to the beginning of the next (during mitosis), in agreement with previous observations that, unlike most pioneer factors, Zld is not mitotically retained on the chromosomes (Dufourt et al., 2018).

RICS analysis also allowed us to determine $\phi$, the fraction of Zld that is immobile (or nearly so) due to either binding to immobile structures (such as DNA) or to forming large aggregates with very low diffusivity (Eqn 4). The immobile fraction of Zld remains nearly the same from nc 10 to nc 13, after which it reaches a steady state at a lower value of ~0.5 at nc 14 (Fig. 2C). Fifty percent of the Zld population was found to be immobile in single-molecule imaging (Mir et al., 2018).

In each nc, at the beginning of interphase, the immobile concentration of Zld, calculated as the product of total nuclear concentration and the immobile fraction, increases, perhaps because the chromatin decondenses and Zld binds the replicated DNA (Antonin and Neumann, 2016; Blythe and Wieschaus, 2016; McDaniel et al., 2019; Yuan et al., 2014), and it starts decreasing after reaching a maximum as chromatin starts condensing before mitosis (Fig. S1A). This is consistent with the rapid formation of dynamic hubs of Zld after mitosis (Mir et al., 2018), the hubs likely being a population included in our immobile concentration measurements. It should be noted that the presence of hubs results in Zld particles with higher brightness than the Zld monomer. As such, the presence of hubs, in principle, could alter our measurements of total Zld nuclear concentration and of the immobile fraction. We have quantified the average particle brightness and found it to be roughly constant over time, suggesting that hubs of Zld have little effect on our measurements (see Materials and Methods and Fig. S2).

We also used cross-correlation RICS (ccRICS) to determine $\psi$, the fraction of the pioneer factors that correlate with His2Av-RFP (Eqn 9 in the Materials and Methods). In *Drosophila*, the enrichment of the histone variant His2Av has been found to correlate with transcriptional potential (Leach et al., 2000; Mavrich et al., 2008), and therefore, the pool correlated with His2Av-RFP (hereafter referred to as the active pool) is likely responsible for pioneering gene activation. As the immobile fraction always measured as larger than the active fraction (Fig. 2C,D), we inferred that the immobile fraction must be composed of two pools: one that correlates with His2Av-RFP and one that does not. Hereafter, the pool that does not correlate with His2Av but is immobile, either due

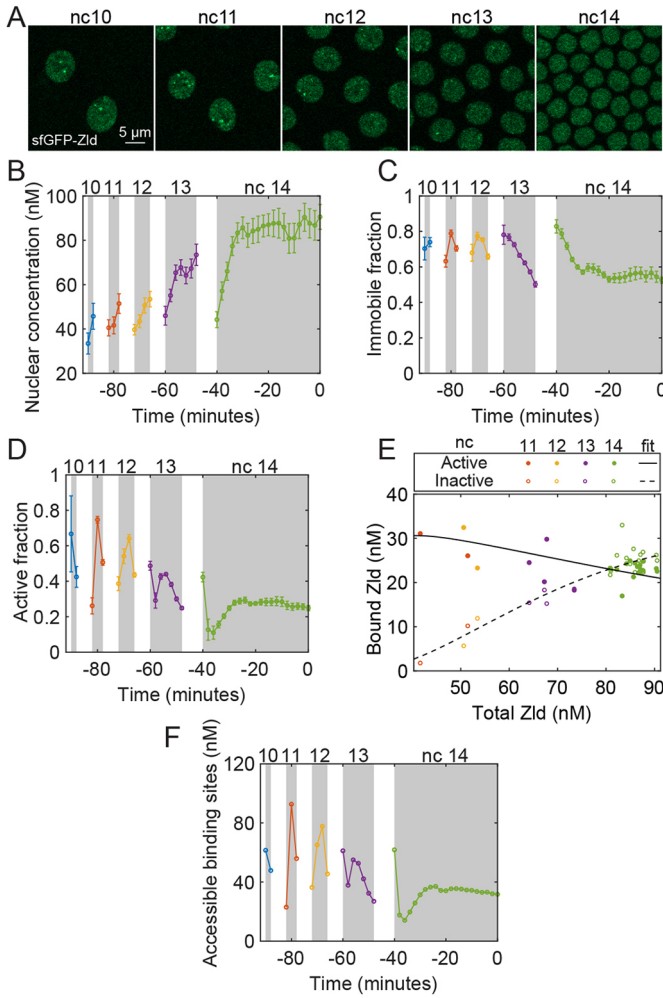

**Fig. 2. Quantification of the biophysical parameters and dynamics of Zld.** (A) Representative images of a sfGFP-Zld embryo used for Raster Image Correlation Spectroscopy (RICS) analysis from nc 10 to nc 14, as indicated. (B-D) Dynamics of different pools of sfGFP-Zld from nc 10 until gastrulation, including total nuclear concentration (B), immobile fraction (C) and active fraction (D). Data are mean±s.e.m. ($n$=20 embryos). (E) Dose/response map between total nuclear concentration of Zld and the immobile (both active and inactive) concentration of Zld. The solid line represents the model fit for the active population and the dashed line represents the same for the inactive population. (F) Change in Zld sites accessible for binding over time. See also Fig. S1 and Movie 1.

to binding to inactive chromatin regions or due to low diffusivity from the formation of large clusters, is referred to as the inactive pool. We found the active fraction decreases from nuclear cycle to nuclear cycle (Fig. 2D). Consequently, the active concentration of Zld, which is the product of the total nuclear concentration and the active fraction of Zld, also decreases, albeit weakly, due to the increasing total nuclear concentration (Fig. S1B). Our results agree with the slight reduction observed in the number of Zld peaks from nc 13 to nc 14 in ChIP-Seq (Harrison et al., 2011). The Zld ChIP-Seq peaks likely represent the binding in the active regions as tightly packed inactive regions are less likely to be identified (Nakato and Shirahige, 2016). Roughly 5 min into nc 14, the concentration of active Zld reaches a steady value around 20-25 nM (Fig. S1B).

Our measurements lend themselves to a general analysis in which the relationship between Zld total nuclear concentration and its binding could be constructed. According to standard thermodynamic equilibrium binding models (see Materials and Methods), an increase in the total Zld concentration should lead to an increase in the bound Zld concentration. However, our data suggest that the active concentration of Zld is a weakly decreasing function of total Zld (Fig. 2E) in which nc 10-12 has a high active concentration of Zld despite the low total Zld concentration, while in nc 14, there is a high total Zld concentration and a slightly lower active concentration. As such, our results suggest that Zld binding is not influenced by Zld concentration alone; additional factors, such as chromatin structure, must be taken into account. Using our data and ChIP-seq results reported by Harrison et al. (2011), the dissociation constant for the active population ($K_{D,a}$) for Zld can be estimated to be ∼17 nM (see Materials and Methods). Fitting a Hill function to our data suggests that the Zld sites accessible for binding reduce over time (Fig. 2F), which might result from the chromatin becoming more densely packed during the later cycles (Harrison et al., 2011; Li et al., 2014; Lowenhaupt et al., 1983). On the other hand, the inactive concentration of Zld increases with time (Fig. S1C), suggesting the inactive fraction may result from binding that is saturating in nature. Fitting a Hill function with a Hill coefficient of 2, due to the sigmoidal nature of the data points, yielded the dissociation constant for the inactive population ($K_{D,i}$) of ∼24 nM and a max concentration of 34 nM (Fig. 2E). These values of $K_D$ suggest that the binding of Zld to DNA or to immobile structures including aggregates is moderately strong. Overall, the dose/response relationship between total and bound Zld concentration is consistent with the previously observed hubs of Zld bound to DNA. Zld molecules within these hubs have short residence times on DNA ($\tau$∼5 s; Mir et al., 2018). This is lower than our estimation (see Materials and Methods), which might result from the high local concentration of Zld within the hub (Mir et al., 2018) or from the local Zld diffusivity (Mirny et al., 2009).

Our data suggest that Zld may not retain its ability to pioneer chromatin accessibility throughout ZGA. Consistent with this, it was recently shown that, in larval type II neuroblasts, Zld binding is influenced by chromatin accessibility (Larson et al., 2021). Thus, our observation may imply that Zld primarily regulates transcription during the minor wave. Indeed, the removal of Zld activity affects early gene expression patterns, resulting in a delay in, but not complete loss of, the transcription of patterning genes (Liang et al., 2008; Nien et al., 2011). Therefore, our results suggest that, during the minor wave of ZGA (nc 10-13), a high concentration of bound Zld might be needed to facilitate the binding of patterning factors present at low levels. On the other hand, during the major wave at nc 14, either the high level of patterning factors, or the presence of other pioneer-like factors, may be sufficient to drive the patterning factor binding and target gene expression when the bound Zld levels have decreased. This raises the question of whether other factors are present during the major wave that may continue to facilitate the binding of other developmental TFs.

## GAF levels increase suddenly in nc 14

GAF, which is encoded by the *Trithorax-like* (*Trl*) gene, plays an essential role in ZGA, along with Zld (Farkas et al., 1994; Gaskill et al., 2021). GAF motifs along with Zld motifs are enriched in the highly occupied target (HOT) regions characterized by open chromatin and bound by many TFs (The modENCODE Consortium et al., 2011; Kvon et al., 2012; Slattery et al., 2014). GAF possesses the properties of a pioneer factor, such as the ability to bind nucleosomal DNA and to create regions of chromatin accessibility by functioning with chromatin remodelers to facilitate the binding of other TFs (Fuda et al., 2015; Gaskill et al., 2021; Judd et al.,

**DEVELOPMENT**

2021; Moshe and Kaplan, 2017; Tang et al., 2022; Tsukiyama et al., 1994).

To measure the dynamics of the nuclear concentration of GAF over time, we performed RICS analysis on the nuclei of blastoderm-stage embryos expressing GAF-sfGFP (Gaskill et al., 2021) (Fig. 3A and Movie 2). The results suggest that the total nuclear concentration of GAF remained nearly constant and very low (∼5-10 nM) from nc 10-13, then showed a sudden increase during nc 14 to ∼30 nM (Fig. 3B). We saw that the nuclear concentration at the beginning of an interphase was similar to that at the end of the previous interphase (Fig. 3B). This observation is consistent with previous work showing GAF is mitotically retained on the chromosomes (Gaskill et al., 2021), and is in contrast to Zld (Fig. 2B), which is not mitotically retained (Dufourt et al., 2018).

The immobile fraction of GAF remained nearly the same during nc 10-13, then decreased slightly during nc 14 (Fig. 3C). The sudden increase in GAF total nuclear concentration, together with only a slight decrease in its immobile fraction, resulted in an increase in the immobile concentration of GAF during nc 14 (Fig. S3A). Although the active fraction of GAF, the pool expected to be bound near the actively transcribed genes, remained approximately the same, on average, from nc 10-14, it varied significantly within each nc (Fig. 3D).

As with Zld, we sought to use our data to map the relationship between GAF nuclear concentration and binding. We noted that, unlike Zld, both active and inactive concentrations of GAF increase with an increase in the total concentration of GAF (Fig. 3E). Our

results agree with the increase in GAF ChIP-seq peaks from nc 9 to 14 observed previously (Gaskill et al., 2021). Both pools appeared to be saturating functions of total GAF concentration, with the inactive pool saturating quickly at a low overall concentration (Fig. 3E and Fig. S3B,C). In contrast, the active pool required higher levels of total GAF to saturate and had a higher capacity. Fitting Hill functions to the data bore out these observations: the inactive pool had a $K_{D,i}$ of 0.4 nM and a max concentration of 9 nM, while the active pool had a $K_{D,a}$ of 5 nM and a max concentration of 35 nM. The $K_D$ values represent a strong affinity of GAF for the binding sites, roughly 4-fold and 60-fold greater than for Zld, consistent with previous reports of stable GAF/DNA binding (Bellec et al., 2022; Espinás et al., 1999; Tang et al., 2022). It appears that, as GAF concentration slowly increases, the majority is apportioned to the inactive pool due to its high affinity. However, because of the low capacity of the inactive pool, upon entering nc 14, the active pool is suddenly able to increase. If the active pool corresponds to transcriptionally active regions of the DNA (i.e. euchromatin), the relative affinities and capacities ensure that GAF acts as a pioneer factor solely during the later, major, wave of ZGA. Furthermore, it has been shown that GAF associates with the GA/CT-rich repeats in the heterochromatin regions throughout cell cycles (Gaskill et al., 2023; Raff et al., 1994), potentially driving transcriptional silencing and the euchromatin association activation during ZGA (Gaskill et al., 2023). As such, the inactive pool may represent GAF bound to heterochromatin. As with Zld, GAF hubs with higher brightness could affect our measurements. Quantification of the brightness suggested small corrections to our data, which did not affect our overall conclusions (see Materials and Methods and Fig. S4).

Our measurements of the biophysical parameters of the pioneer factors and their dynamics allowed the construction of input/output maps between total concentration of the factors and their DNA binding. While a subset of the pioneer factor bound regions has been found to stay inaccessible (Freund et al., 2024; Gibson et al., 2024), binding of pioneer factors to DNA is an indispensable step for establishing and maintaining chromatin accessibility. Following this, binding of a correct combination of TFs can lead to the activation of the accessible regions (Brennan et al., 2023). Using the input/output maps, we can infer that Zld acts as a global activator of early genes enabled by the high active concentration of Zld during the minor wave of ZGA; GAF comes into play during the major wave, as indicated by the high active concentration of GAF during this time (Fig. 4). This is supported by the fact that the regions that gain accessibility early during ZGA are enriched for Zld binding, whereas those that gain accessibility late are enriched for GAF binding (Blythe and Wieschaus, 2016). A similar quantitative approach, in which accurate measurements of biophysical parameters of TFs and their dynamics enable the construction of input/output map between TF concentration and transcriptional activity, is expected to aid other gene expression studies.

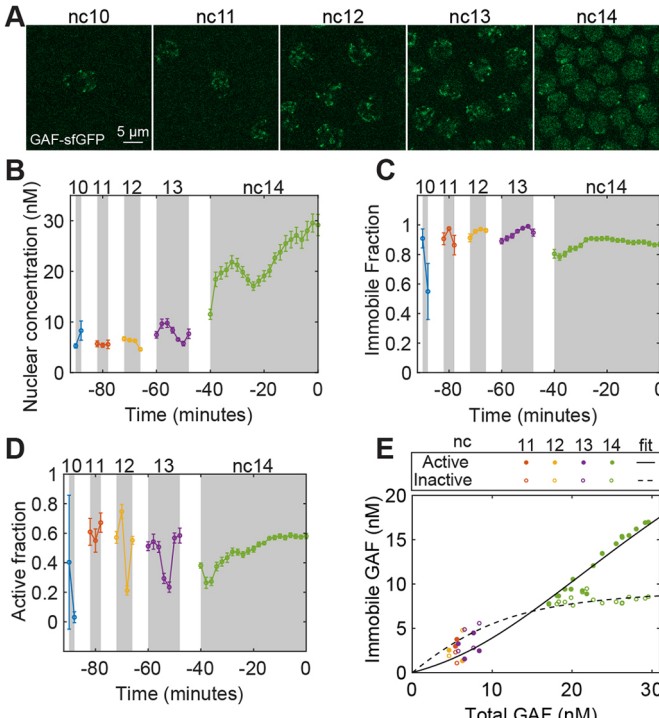

**Fig. 3. Quantification of the biophysical parameters and dynamics of GAF.** (A) Representative images of a GAF-sfGFP(C) embryo used for Raster Image Correlation Spectroscopy (RICS) analysis from nc 10 to nc 14, as indicated. (B-D) Dynamics of the parameters from nc 10 until gastrulation, including total nuclear concentration (B), the immobile fraction (C) and the active fraction (D). Data are mean±s.e.m. (*n*=19 embryos). (E) Dose/response map between total nuclear concentration of GAF and the immobile (both active and inactive) concentration of GAF. The solid line represents the model fit for the active population and the dashed line represents the same for the inactive population. See also Fig. S3 and Movie 2.

## MATERIALS AND METHODS
### *Drosophila* strains
The fly strain used for the Zld imaging is *sfGFP-zld; His2Av-RFP (II)*. The *sfGFP-zld* mutant allele was generated using Cas9-mediated genome engineering by Hamm et al. (2017). The fly strain used for the GAF imaging is *His2Av-RFP (II); GAF-sfGFP(C) (III)*. The *GAF-sfGFP(C) (III)* mutant allele was generated using Cas9-mediated genome engineering by Gaskill et al. (2021).

### Sample preparation
Flies were raised on standard cornmeal-molasses-yeast medium at 25°C. Fly cages were prepared with desired fly strains and kept at room temperature for

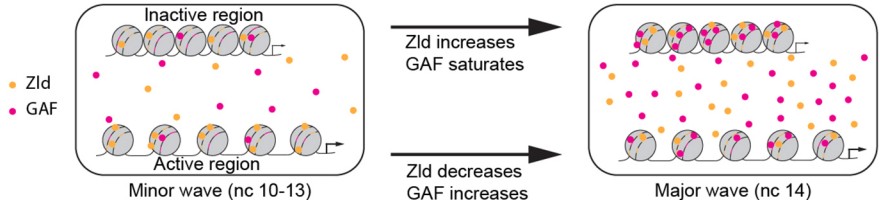

**Fig. 4. Role of pioneer factors Zld and GAF during *Drosophila* ZGA.** During the minor wave of ZGA, a high concentration of Zld (yellow circles) bound to its sites (yellow lines) in the active regions allows it to act as the main activator of early genes. This active bound concentration reduces during the major wave, when the concentration of GAF (magenta circles) bound to the sites (magenta lines) in the active regions increases, after saturating the sites in the inactive regions. This allows GAF to act as the main activator of the genes that begin to be expressed during the major wave.

2 days before imaging. Grape juice agar plates streaked with yeast paste were placed onto the bottoms of the cages for oviposition. For imaging, the flies were allowed to lay eggs for 1 h after which the plates were removed for embryo collection. The embryos were transferred from the plates to mesh baskets, dechorionated using bleach for 30 s and washed with deionized water to remove residual bleach (Carrell and Reeves, 2015).

### Live imaging

After dechorionation, the embryos were mounted in 1% low melting point agarose (IBI Scientific, IB70051) in deionized water on a glass bottom Petri dish (MatTek, P35G-1.5-20-C) and deionized water was poured into the Petri dish over the solidified low melting point agarose to cover the samples. The low melting point agarose rendered mechanical stability to the embryos while remaining transparent submerged in water (Huisken and Stainier, 2009; Kaufmann et al., 2012). All images were collected on a Zeiss LSM 900 confocal laser scanning microscope. For the image acquisitions, C-Apochromat 40×/1.2 water immersion Korr objective, 488 nm laser for sfGFP, 561 nm laser for RFP and GaAsP-PMT detector were used. The detector was operated at 650 V with 1× gain and 0% offset. Emission was detected in the range of 410-546 nm for sfGFP and 595-700 nm for RFP. All the images had 1024×1024 pixel resolution. The images were collected at 5× zoom, resulting in pixel size of 31.95 nm and a pixel dwell time of 2.06 μs. This corresponds to a frame time of 5.06 s and a line time of ∼5 ms (ratio of frame time and number of rows in the image). Image acquisition was started when the embryos were at nc 10 and continued until gastrulation, as indicated by the nuclear morphology.

### Raster image correlation spectroscopy image analysis

Raster image correlation spectroscopy (RICS) analysis, a derivative of fluorescence correlation spectroscopy, entails constructing autocorrelation functions (ACFs) from imaging data and fitting ACF models to these data-derived ACFs. We performed these analyses according to previous protocols (Al Asafen et al., 2024; Schloop et al., 2025). In brief, live imaging time courses were background subtracted and divided into groups of 7-12 frames. The frames within a group were averaged together to create an averaged frame for that group, which was used for two purposes. First, the nuclei in the averaged frame were segmented according to a watershed algorithm. This segmented nuclear mask was used for each frame in the group. Second, the immobile variation within each frame was then removed by subtracting by the averaged frame on a pixel-by-pixel basis, then the scalar average intensity value of the averaged frame was added back (Digman et al., 2005a,b). Two dimensional (2D) RICS autocorrelation functions of the nuclear fraction of each frame were built using a fast Fourier transform protocol in Matlab, and these ACFs were averaged together for all frames in a given group. The result was a time series of 2D ACFs, each of which corresponded to a given grouping of 7-12 frames. Background subtraction was performed on the fly by examining the histogram of intensities and fitting a Gaussian to the lowest intensity pixels in the image.

### Experimental determination of axial displacement

We mounted diffraction-limited TetraSpeck beads (0.1 μm, T7279, Invitrogen) in 1% low melting point agarose on a glass bottom Petri dish following the same protocol as the embryos. Similarly, the agarose was immersed in deionized water upon solidification. The microscope parameters, such as the pixel size and frame time, were identical as the RICS acquisitions. We acquired z-stacks with 0.05 μm distance between the slices. We acquired six z-stacks: three different locations in the agarose and two different batches of beads mounted in the agarose.

The beads were then detected in the average image of the z-stack. A 3D Gaussian was fitted to the intensity of the beads using fmincon. The centroid coordinates of each detected bead and the highest intensity z-plane for the corresponding bead were used as the initial guesses for the centers in x, y and z directions. This was carried out separately for the two channels (Fig. S5A,B). The difference between the centers (x, y and z) for the two channels, as determined from the fit, were calculated for each bead (Fig. S5C-D). The full width at half maximum (FWHM) in the z-direction was calculated separately for the two channels (Fig. S5E).

### Fitting RICS autocorrelation functions to estimate concentration and mobility

RICS autocorrelation functions (ACFs) generally contain two orthogonal pieces of information: the amplitude determines the concentration of the species and the shape determines the mobility. For a purely diffusing species, the theoretical, normalized ACF, $\hat{G}(\Delta x, \Delta y)$, is the following:

$$
\hat{G}(\Delta x, \Delta y) = \left( \frac{1}{1 + \frac{4D(\tau_p \Delta x + \tau_\ell \Delta y)}{w_0^2}} \right) \sqrt{\frac{1}{1 + \frac{4D(\tau_p \Delta x + \tau_\ell \Delta y)}{w_z^2}}}
$$
$$
\times \exp \left( -\frac{(\Delta x^2 + \Delta y^2) \Delta r^2}{w_0^2 + 4D(\tau_p \Delta x + \tau_\ell \Delta y)} \right),
\tag{1}
$$

where $D$ is the diffusivity. This equation has five microscope parameters: $w_0$ and $w_z$ are the radii of the point spread function (PSF) in the $xy$ plane and the axial ($z$) direction, respectively; $\Delta r$ is the $xy$ size of a pixel; and $\tau_p$ and $\tau_\ell$ are the pixel dwell time (determined by the scan speed) and line time (determined by a combination of the scan speed and number of pixels in the width of the image), respectively. The two independent variables, $\Delta x$ and $\Delta y$, are the pixel shifts in the fast and slow directions, respectively.

The non-normalized ACF includes the amplitude $A$, which is equal to the reciprocal of $\bar{n}$ (the average number of molecules in the confocal volume):

$$
G(\Delta x, \Delta y) = A\hat{G}(\Delta x, \Delta y).
\tag{2}
$$

In practice, $A = \gamma/\bar{n}$, where $\gamma = \sqrt{2}/4$ is a factor that accounts for the uneven illumination airy unit (Brown et al., 2008). Note that $\bar{n}$ can be converted to $\bar{c}$, the average concentration in the confocal volume, $V_{PSF}$, by $\bar{n} = V_{PSF} \bar{c}$, where $V_{PSF} = \pi^{\frac{3}{2}} w_0^2 w_z$.

The 2D ACFs for sfGFP-Zld or GAF-sfGFP for each time point were then used to fit two different models. First, the fast direction cut of the 2D ACF, $G_s(\Delta x, 0)$, was used to fit a Gaussian equation that approximates the PSF of the microscope. Because $\Delta y = 0$ along this cut, and $4D\tau_p \Delta x/w_z^2 \ll 1$, Eqns 1

and 2 simplify to:

$$G(\Delta x, 0) \approx (A - B) \exp\left(-\frac{\Delta x^2 \Delta r^2}{w_0^2}\right) + B, \tag{3}$$

where, for robustness of fit, a small adjustable background constant, $B$, was added, and $w_0$ was allowed to vary slightly. To avoid problems with background, $B$ was constrained to have a magnitude less than $10^{-3}$; in practice, it never exceeded 1% of the value of the amplitude, $A$. This first fitting step resulted in accurate estimates of the ACF amplitude, $A$.

Second, holding $A$ fixed, the entire 2D ACF was then used to fit a two component model, $G_{2c}(\Delta x, \Delta y)$, which is a linear combination of a freely diffusing fraction and an immobile fraction (Al Asafen et al., 2024):

$$\hat{G}_{2c}(\Delta x, \Delta y) = \phi\hat{G}_{bind}(\Delta x, \Delta y) + (1 - \phi)\hat{G}_{diff}(\Delta x, \Delta y), \tag{4}$$

where $\hat{G}_i(\Delta x, \Delta y)$ is given by Eqn 1 [with $\hat{G}_{bind}(\Delta x, \Delta y)$ having $D=0$ and $\hat{G}_{diff}(\Delta x, \Delta y)$ having $D$ non-zero; see Fig. 1D], the linear combination weight, $\phi$, is the immobile fraction and $\hat{G}_{2c}$ is the normalized two-component ACF, such that:

$$G_{2c}(\Delta x, \Delta y) = (A - B)\hat{G}_{2c}(\Delta x, \Delta y) + B. \tag{5}$$

The parameter $B$ in Eqn 5 may have a different value from the one found in Eqn 3. The 2D ACF for His2Av-RFP for each time point was used to fit only the 2D PSF to obtain a measure of the ACF amplitude.

### Fitting ccRICS cross-correlation functions to estimate correlated binding

Like the 2D ACF, the 2D cross-correlation function (CCF) between either sfGFP-Zld or GAF-sfGFP and His2Av-RFP was computed through a fast Fourier transform protocol in Matlab. This 2D CCF was then used to fit a 2D model of cross correlation (Al Asafen et al., 2024):

$$G_{cc}(\Delta x, \Delta y) = (A_{cc} - B)\exp\left(-\frac{(\Delta x \Delta r - d_x)^2 + (\Delta y \Delta r - d_y)^2}{\hat{w}_0^2} - \frac{d_z^2}{\hat{w}_z^2}\right) + B. \tag{6}$$

Because cross correlation uses two lasers, the two PSFs are different in size and their centers are generally not concurrent. In Eqn 6, the parameters $d_x, d_y$ and $d_z$ represent the displacements between the centers of the two PSFs in the $x$, $y$ and $z$ directions, respectively. The two displacements in the $xy$ plane are adjustable parameters that can be estimated from the 2D CCF. The axial displacement was estimated by imaging fluorescent beads (see above), and our measurements resulted in the factor $\exp(d_z^2/\hat{w}_z^2)$ being roughly equal to 1.02, implying the factor could be safely ignored. As with the ACF, for robustness of fit, a small adjustable background constant, $B$, was added. Fitting Eqn 6 to the CCF data resulted in an estimate of the CCF amplitude, which is defined as:

$$A_{cc} = \frac{\bar{c}_2}{\pi^{3/2}\hat{w}_0^2\hat{w}_z(\bar{c}_1 + \bar{c}_2)(\bar{c}_2 + \bar{c}_3)}, \tag{7}$$

where the average PSF sizes are defined as $\hat{w}_0^2 = 0.5(w_{0,g}^2 + w_{0,r}^2)$ and $\hat{w}_z^2 = 0.5(w_{z,g}^2 + w_{z,r}^2)$, and the subscripts $g$ and $r$ denote the green and red channels, respectively, and where $\bar{c}_1$ is the average concentration of the sfGFP-containing species that does not cross-correlate with the RFP-containing species (in the rest of the paper, this is the free pool plus the inactive pool); $\bar{c}_2$ is the average concentration of the active pool and $\bar{c}_3$ is the average concentration of the RFP-containing species that does not cross-correlate with the sfGFP-containing species, which include DNA-bound molecules, but could also include freely diffusing molecules.

To obtain $\psi$, which is the fraction of sfGFP-Zld or GAF-sfGFP correlated to His2Av-RFP, we calculate the ACF amplitude of the red channel, $G_{red}(\Delta x, \Delta y)$, then use it to fit Eqn 3, yielding the amplitude $G_{red}(0, 0) = A_{red} = 1/(\pi^{3/2}w_{0,r}^2 w_{z,r}(\bar{c}_2 + \bar{c}_3))$. The ratio of $A_{cc}$ to $A_{red}$ then

gives:

$$\frac{A_{cc}}{A_{red}} = \frac{w_{0,r}^2 w_{z,r}}{\hat{w}_0^2\hat{w}_z}\frac{\bar{c}_2}{\bar{c}_1 + \bar{c}_2} = \frac{w_{0,r}^2 w_{z,r}}{\hat{w}_0^2\hat{w}_z}\psi. \tag{8}$$

Therefore, we can calculate $\psi$ as:

$$\psi = \frac{\hat{w}_0^2\hat{w}_z}{w_{0,r}^2 w_{z,r}}\frac{A_{cc}}{A_{red}}. \tag{9}$$

### Thermodynamic equilibrium model

When the binding of a TF to DNA is in equilibrium, the probability of the TF binding can be modeled as described by Bintu et al. (2005):

$$P\{TF\ bound\} = \frac{C_{free}^n}{C_{free}^n + K_D^n}, \tag{10}$$

where $C_{free}$ is the nuclear concentration of the free TF, $K_D$ is the dissociation constant, $P\{TF\ bound\}=C_{bound}/C_B$, $C_{bound}$ is the concentration of the bound TF and $C_B$ is the concentration of binding sites. A Hill coefficient of $n=1$ was used in all the cases unless mentioned otherwise.

The equation was solved and adjustable parameters (dissociation constant and binding site concentrations) were determined using the least-squares fitting algorithm lsqcurvefit and the global optimization solver MultiStart in Matlab.

To express the concentration of active or inactive bound TF as a function of total nuclear concentration, we used the conservation relationship between different pools of the TF:

$$C_{tot} = C_{free} + C_{active} + C_{inactive}, \tag{11}$$

where $C_{tot}$ is the total nuclear concentration, $C_{active}$ is the concentration of the active pool and $C_{inactive}$ is the concentration of the inactive pool.

Using Eqn (10) this can be written as:

$$C_{tot} = C_{free} + \frac{C_{free}C_{B_a}}{C_{free} + K_{D_a}} + \frac{C_{free}C_{B_i}}{C_{free} + K_{D_i}}, \tag{12}$$

where $C_{B_a}$ and $C_{B_i}$ are the concentrations of binding sites for the active and inactive pools, respectively. $K_{D_a}$ and $K_{D_i}$ are the dissociation constants for the active and inactive pools, respectively.

In the case of Zld, $C_{B_a}$ can be expressed as a function of $C_{free}$ (see Fig. S1D). Empirically, we modeled this relationship as a power law:

$$C_{B_a} = aC_{free}^{-k}, \tag{13}$$

where $a=420$ and $k=0.7$ were the best-fit parameters (see Fig. S1D). The value of $C_{B_a}$ from Eqn 13 can be substituted into Eqn 12. Eqn 12 can then be used to determine the concentrations of free TF corresponding to a given total concentration to construct the dose/response maps in Figs 2E and 3E. For these calculations, nc 10 is ignored due to its short duration and data points near the start of each nc are excluded due to equilibrium not being established during these time points.

### Zld accessible sites

Pooled data from nc 8, 13 and 14 identified 12,135 Zld ChIP-Seq peaks (Harrison et al., 2011). We assumed that, during the early ncs, the chromatin is fairly open and all the ChIP-Seq identified sites are accessible for binding, such that the number of Zld sites accessible for binding during nc 11 is also assumed to be 12,135. Given that the nuclear volume during nc 11 is roughly 435 μm$^3$ (Foe and Alberts, 1985), the concentration of Zld sites accessible for binding, averaged over the whole nucleus, is roughly, L = $\frac{12{,}135\ \text{peaks}}{435\ \text{μm}^3}$ = 46.34 nM. However, given that *Drosophila* is diploid, we set L=2×46.34=92.68 nM. Using these estimates, Eqn 10 and our data for nc 11, we arrive at $K_{D,a}$=17 nM. Using this $K_D$ and our data in Eqn 10, the change in Zld sites accessible for binding over time was estimated.

### Residence time of Zld

Dissociation constants are typically defined as $K_D = \frac{k_{off}}{k_{on}}$; where $k_{on}$ and $k_{off}$ are the on and off rates for the TF-immobile structure interaction. For

protein-ligand binding events in which the on-rate is limited by three dimensional diffusion, $10^{-4}$ nM$^{-1}$ s$^{-1}$<$k_{on}$<$10^{-3}$ nM$^{-1}$ s$^{-1}$ (Wittrup et al., 2019). Using this rule of thumb for the $k_{on}$ value and $K_D$~20 nM (valid for both the active and inactive pools of Zld), the residence time, $\tau = \frac{1}{k_{off}} = \frac{1}{k_{on}K_D}$, can be estimated to be $\tau$~8 min−50 s. However, if the diffusivity is higher, as would be the case of one-dimensional sliding along the DNA (Mirny et al., 2009), $k_{on}$ is closer to $10^{-1}$ nM$^{-1}$ s$^{-1}$, resulting in $\tau$~0.5 s. Owing to this variability using the rule of thumb, we used the Smoluchowski equation (Mirny et al., 2009) for the Zld-DNA interaction. The diffusivity of Zld obtained from our analysis (D) ~2 μm$^2$/s. According to the Smoluchowski equation (Mirny et al., 2009): $k_{on}=4\pi Dba$, where cross-section of the binding reaction (b)=0.34 nm and the fraction of the molecular surface of the protein that contains the reactive binding interface (a) ~0.2-0.5. Therefore, $k_{on}$~0.001-0.003 nM$^{-1}$ s$^{-1}$, resulting in $\tau$~19−49 s, which agrees with the lower limit of the value calculated using the rule of thumb.

## Brightness analysis

The average molecular brightness, $Q$, of particles in the nucleus was calculated in the following manner. First, the average apparent brightness, $B$, of the sfGFP-containing particles was calculated as the ratio of the variance, $\sigma^2$, of the intensity of pixels in the nuclei divided by their intensity, $I$:

$$B = \frac{\sigma^2 - \sigma_0^2}{I},$$

where $\sigma_0^2$ is the variance of a zero intensity image, which can also be found as the variance of the lowest intensity pixels in the image. Plots of $B$ over nc 10-14 can be found in Figs S2A (Zld) and S4A (GAF).

Next, the $S$-factor, which relates apparent brightness and molecular brightness, was calculated as the slope of the best-fit line relating the microscope shot noise, $\sigma_{shot}^2$ (which is found in the pixels in the nuclei) to the intensity of the nuclei (Dalal et al., 2008):

$$\sigma_{shot}^2 = SI.$$

In practice, the shot noise is equal to the RICS ACF amplitude times the intensity squared, $AI^2$, subtracted from the variance described above, $\sigma^2 - \sigma_0^2$. The shot noise and intensity at each time point in the time course was used to compute the best-fit line (see Fig. S2B for Zld and S4B for GAF). The equation used was $\sigma_{shot}^2 = SI + b$, where $b$ is a free parameter added for robustness of fit and was found generally to be close to zero. For Zld, $S$=363; for GAF, $S$=413.

Next, to convert the average apparent brightness into the average molecular brightness, the following relationship was used (Dalal et al., 2008):

$$Q = \frac{B}{S} - 1.$$

The plots of $Q$ over nc 10-14 can be found in Figs S2C (Zld) and S4C (GAF).

Corrections for changes in brightness for GAF were performed in the following manner. First, a minimum value of $Q$ was determined ($Q_{min}$) and the brightness time course was normalized by $Q_{min}$: $q(t)=Q(t)/Q_{min}$, with values of $q(t)$<1 being set to 1. Next, new values of the RICS ACF amplitude, $A_{new}$, and of the immobile fraction, $\phi_{new}$, were calculated from $q$ and the old values ($A_{old}$ and $\phi_{old}$):

$$A_{new} = \frac{A_{old}}{q}$$

$$\phi_{new} = 1 - q(1 - \phi_{old}).$$

From these, corrected values of all concentrations, the free fraction and the uncorrelated fraction were calculated. These corrections were performed at two levels. In the first level, we assumed all variations within nc 14 were within the range of being roughly constant. In other words, only variations outside of this range were corrected for, meaning that $Q_{min}$ was set as the maximum value observed during nc 14 (see Fig. S4C). This resulted in corrections made for the increase in brightness in the final point of nc 12 and

the final four points of nc 13. While these corrections had a quantitative effect on our results, the corrected relationships between bound and free GAF are maintained (Fig. S4D,E).

### Acknowledgements
We thank Melissa M. Harrison for kindly providing fly stocks and for helpful discussion of the manuscript.

### Competing interests
The authors declare no competing or financial interests.

### Author contributions
Conceptualization: S.S.D., G.T.R.; Data curation: S.S.D.; Formal analysis: S.S.D., G.T.R.; Funding acquisition: G.T.R.; Investigation: S.S.D., G.T.R.; Methodology: S.S.D., G.T.R.; Project administration: G.T.R.; Resources: G.T.R.; Software: G.T.R.; Supervision: G.T.R.; Validation: S.S.D.; Visualization: S.S.D., G.T.R.; Writing – original draft: S.S.D., G.T.R.; Writing – review & editing: S.S.D., G.T.R.

### Funding
This work was supported in part by a National Science Foundation grant (MCB-2105619). Open Access funding provided by Texas A&M University. Deposited in PMC for immediate release.

### Data and resource availability
The image analysis pipeline and code are available on Zenodo (10.5281/zenodo.15701860). All the image files have been deposited in the Texas Data Repository (10.18738/T8/B4A3SJ).

### Peer review history
The peer review history is available online at https://journals.biologists.com/dev/lookup/doi/10.1242/dev.204460.reviewer-comments.pdf

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
