## [Peer Review File · Development (Cambridge, England)]

Bulk-level maps of pioneer factor binding dynamics during *Drosophila* maternal-to-zygotic transition

Sadia Siddika Dima and Gregory T. Reeves
DOI: 10.1242/dev.204460

Editor: Paul Francois

Review timeline

Original submission:	16 October 2024
Editorial decision:	13 January 2025
First revision received:	27 March 2025
Editorial decision:	12 May 2025
Second revision received:	22 May 2025
Accepted:	June 5 2025

Original submission

First decision letter

MS ID#: dev.204460

MS TITLE: Bulk-level maps of pioneer factor binding dynamics during *Drosophila* maternal-to-zygotic transition

AUTHORS: Sadia Siddika Dima and Gregory T. Reeves

Dear Dr Reeves,

I have now received all the referees' reports on the above manuscript, and have reached a decision. The referees' comments are appended below, or you can access them online: please go to:

As you will see, the referees express considerable interest in your work, but have some significant criticisms and recommend a substantial revision of your manuscript before we can consider publication. If you are able to revise the manuscript along the lines suggested, which may involve further experiments, I will be happy receive a revised version of the manuscript. Your revised paper will be re-reviewed by one or more of the original referees, and acceptance of your manuscript will depend on your addressing satisfactorily the reviewers' major concerns. Please also note that Development will normally permit only one round of major revision. If it would be helpful, you are welcome to contact us to discuss your revision in greater detail. Please send us a point-by-point response indicating your plans for addressing the referees' comments, and we will look over this and provide further guidance.

Please attend to all of the reviewers' comments and ensure that you clearly highlight all changes made in the revised manuscript. Please avoid using 'Tracked changes' in Word files as these are lost in PDF conversion. I should be grateful if you would also provide a point-by-point response detailing how you have dealt with the points raised by the reviewers in the 'Response to Reviewers' box. If you do not agree with any of their criticisms or suggestions please explain clearly why this is so.

Reviewer 1

Advance summary and potential significance to field

The authors report on their study of two pioneer transcription factors (TFs) active in *Drosophila* embryos, Zelda and GAGA factor. They used raster image correlation spectroscopy to determine, over the course of nuclear cycles 10 to 14: 1) the total nuclear concentration of these TFs (directly calculated from the amplitude of the measured autocorrelation functions), 2) the fraction of the TFs that is immobile, presumably because of association to DNA (interpreting the slow-scanning direction of 2D ACFs as a two-component decay corresponding to an immobile or very slow, and a mobile population) and 3) the fraction of the bound TFs that are bound to active DNA regions as defined by the presence of His2Av-RFP (from the relative amplitude of the cross-correlation function). This allowed the authors to establish titration curves, showing the fraction of TFs bound to either active or inactive DNA regions as a function of TF concentration, and to extract dissociation constants.

The provision of quantitative *in vivo* data related to the binding to DNA of these two important TFs which control the binding of many other TFs brings to light a number of very interesting facts. It clarifies in which time range during zygotic genome activation Zelda and GAGA factor are most active (Zelda during the early minor wave, and GAGA factor during the later major wave). The interpretation of the data also suggests a mechanism for why that is, despite the fact that the overall concentration of both TFs increase over time: the association of GAGA factor with active DNA regions is reinforced at high TF concentration, while the inverse is true for Zelda. This proposition provides a very interesting framework for thinking about the control of transcription by TFs, and morphogens in particular, over space and time. It also calls for understanding what makes Zld and GAGA factor so different with respect to their binding to active DNA.

Comments for the author

The methods employed are appropriate, and the data analysis is detailed and careful. Data interpretation is rational and well thought out, but some simplifying assumptions (some acknowledged, some not) are made along the way, which I think should be mentioned and discussed, so that the accuracy of the measured quantity and the solidity of the proposed model can be fairly assessed by readers.

1) The authors mention that Zelda is known to localize to hubs, and that it would affect the estimated residence time of the TFs on DNA. However, the implication that the formation of these hubs would have on the interpretation of the RICS data, and in particular their concentration measurements, is not discussed. If some of the detected species are oligomers, then the simple relationship between ACF amplitude and fluorophore concentration is no longer valid (Eqs. 1 and 2 are only strictly correct for either a single species, or two species with the exact same molecule brightness). This possibility should be at least acknowledged and discussed. Would it lead to overestimating or underestimating the total TF concentration? Similarly, if the immobile species detected is a oligomeric hub, then the interpretation of ϕ as the fraction of immobile TFs is not longer valid (Eq. 4), as ϕ would then also depend on the ratio between the molecular brightness of both species. Discussing this is important.

2) Background noise can also affect concentration measurements, so the fact that background was subtracted, and that a background term needed to be added to the fit of the ACFs (Eq. 3) could have consequences on the measured value of the concentration. Again, this should be acknowledged and discussed.

3) One way to check that neither oligomers nor background is a concern, and to demonstrate that the signal-to-noise ratio in the experiment is high, is to report on the apparent molecular brightness of the detected fluorescent molecules. If this apparent molecular brightness remains constant over the course of the embryo development, or even over the course of a nuclear cycle, it would be a very good sign that neither oligomerization nor background noise (which might vary over time) is a concern. So what was the molecule brightness in these experiments?

4) When interpreting the cross-correlation measurements, the authors mention that the axial displacement between the two laser foci is predicted to be small, but there is no reference for this statement. In my experience, this axial displacement is not in fact always small, even with a high-

quality apochromat objective. A non-zero axial displacement would lead to an underestimate of the bound fraction. Ideally this displacement would be experimentally determined with a calibration experiment, but at the very least the possibility that it is not zero should be acknowledged. It would also be interesting to report on the measured values of the xy plane displacement.

Additional minor comments.

5) The model proposed by the authors seem to assume that the studied TFs passively bind to either "passive" or "active" DNA regions. But is it possible that they themselves participate in the activation of the DNA?

6) The fit of the 2D ACFs should return not only the value of the TFs immobile fraction, but also the diffusivity of the mobile fraction. Why not report this value, and use it to estimate the limit value for k_{on} (in the "residence time of Zld" section)?

Reviewer 2

Advance summary and potential significance to field

Dima and Reeves estimate the bound and unbound concentration of two critical early transcription factors in the *Drosophila* embryo during zygotic genome activation. The paper employs raster image correlation spectroscopy (RICS) to examine how Zelda-sfGFP and GAF-sfGFP change during zygotic genome activation (ZGA). They find that Zelda concentrations increase over the pre-ZGA cycles, but the amount bound to DNA is more constant. Conversely, they find that GAF binding increases dramatically in NC14. This work represents an orthogonal approach to previous work documenting both Zelda and GAF nuclear behavior through other forms of quantitative imaging and sequencing based approaches. It adds incrementally to our understanding of how these TFs change during ZGA.

Comments for the author

Major:

How these results fit with previous literature should be more thoroughly discussed. ChIP-seq and an abundance of live imaging data already exist for both Zelda and to a lesser extent GAF. How do these findings compare to previous work? Especially about the known behavior of Zelda at transcriptional hubs and the changes in occupancy at specific sites from ChIP-seq data? The changes in total TF nuclear concentration are well documented, so the novelty of this work lies primarily in the measurements of the fraction bound.

While in general the assumption that histones are in chromatin is well justified, in the early embryo there is a vast excess of maternally provided histone. Indeed, the majority of histone H3 is not bound to chromatin in NC11. The assumption that H2Av-RFP marks exclusively chromatin may not hold in the early embryo. The authors should consider how this might affect their interpretation of the ccRICS data.

Minor:

In the abstract and results it would be helpful to be more precise about what period is under consideration (NC10-NC14) when discussing the TF concentration changes. It is clear in the figures, but not specified in the text until very late.

Given the importance of line scan time to the RICS method it should be reported (not just frame time) in the methods.

There is a typo: "This raises the question of whether other factors are present in during the major wave".

First revision

Author response to reviewers' comments

Comments from the Reviewers:

Reviewer 1:**SUMMARY OF THE ADVANCE MADE IN THIS PAPER AND ITS POTENTIAL SIGNIFICANCE TO THE FIELD**

The authors report on their study of two pioneer transcription factors (TFs) active in *Drosophila* embryos, Zelda and GAGA factor. They used raster image correlation spectroscopy to determine, over the course of nuclear cycles 10 to 14: 1) the total nuclear concentration of these TFs (directly calculated from the amplitude of the measured autocorrelation functions), 2) the fraction of the TFs that is immobile, presumably because of association to DNA (interpreting the slow-scanning direction of 2D ACFs as a two-component decay corresponding to an immobile or very slow, and a mobile population) and 3) the fraction of the bound TFs that are bound to active DNA regions as defined by the presence of His2Av-RFP (from the relative amplitude of the cross-correlation function). This allowed the authors to establish titration curves, showing the fraction of TFs bound to either active or inactive DNA regions as a function of TF concentration, and to extract dissociation constants.

The provision of quantitative *in vivo* data related to the binding to DNA of these two important TFs which control the binding of many other TFs brings to light a number of very interesting facts. It clarifies in which time range during zygotic genome activation Zelda and GAGA factor are most active (Zelda during the early minor wave, and GAGA factor during the later major wave). The interpretation of the data also suggests a mechanism for why that is, despite the fact that the overall concentration of both TFs increase over time: the association of GAGA factor with active DNA regions is reinforced at high TF concentration, while the inverse is true for Zelda. This proposition provides a very interesting framework for thinking about the control of transcription by TFs, and morphogens in particular, over space and time. It also calls for understanding what makes Zld and GAGA factor so different with respect to their binding to active DNA.

SUGGESTIONS TO AUTHORS

The methods employed are appropriate, and the data analysis is detailed and careful. Data interpretation is rational and well thought out, but some simplifying assumptions (some acknowledged, some not) are made along the way, which I think should be mentioned and discussed, so that the accuracy of the measured quantity and the solidity of the proposed model can be fairly assessed by readers.

1) The authors mention that Zelda is known to localize to hubs, and that it would affect the estimated residence time of the TFs on DNA. However, the implication that the formation of these hubs would have on the interpretation of the RICS data, and in particular their concentration measurements, is not discussed. If some of the detected species are oligomers, then the simple relationship between ACF amplitude and fluorophore concentration is no longer valid (Eqs. 1 and 2 are only strictly correct for either a single species, or two species with the exact same molecule brightness). This possibility should be at least acknowledged and discussed. Would it lead to overestimating or underestimating the total TF concentration? Similarly, if the immobile species detected is a oligomeric hub, then the interpretation of ϕ as the fraction of immobile TFs is not longer valid (Eq. 4), as ϕ would then also depend on the ratio between the molecular brightness of both species. Discussing this is important.

We thank the reviewer for suggesting this update. We had previously left this effect unexplained because we did not focus on the clusters for this paper. We have investigated the molecular brightness and discuss this further in the reviewer's Point #3.

2) Background noise can also affect concentration measurements, so the fact that background was subtracted, and that a background term needed to be added to the fit of the ACFs (Eq. 3) could have consequences on the measured value of the concentration. Again, this should be acknowledged and discussed.

The background term in the model fits to the ACF was only for robustness of the fit, and, as mentioned in the manuscript, was constrained to be a small value: magnitude no greater than 0.001. In context, the value of B never exceeded 1% of the value of A. We have added these comments to the methods section. See line 337.

3) One way to check that neither oligomers nor background is a concern, and to demonstrate that the signal-to-noise ratio in the experiment is high, is to report on the apparent molecular brightness of the detected fluorescent molecules. If this apparent molecular brightness remains constant over the course of the embryo development, or even over the course of a nuclear cycle, it would be a very good sign that neither oligomerization nor background noise (which might vary over time) is a concern. So what was the molecule brightness in these experiments?

We appreciate this suggestion to quantify the brightness over the time course of nc 10-14. We calculated the apparent brightness as roughly the variance over the intensity of the image. We further corrected this calculation to the molecular brightness by measuring the conversion factor as the slope of the relationship between the shot noise and the intensity of the image (see Dalal et al., 2008). After doing these calculations, we determined that the molecular brightness for Zld is roughly constant across the time course, and we have provided this quantification as a supplemental figure (Figure S2). For GAF, we noticed that the brightness spikes near the end of nc 12 and 13, and we have provided corrections for this observation as a supplemental figure (Figure S4). In addition, there is some small systematic variation in brightness within nc 14 for GAF, and we have provided an additional correction for that observation in a separate figure. Neither correction contravenes our qualitative results for GAF. We have also included a discussion of these calculations in the Methods section (Subsection: Brightness analysis, line 438).

4) When interpreting the cross-correlation measurements, the authors mention that the axial displacement between the two laser foci is predicted to be small, but there is no reference for this statement. In my experience, this axial displacement is not in fact always small, even with a high-quality apochromat objective. A non-zero axial displacement would lead to an underestimate of the bound fraction. Ideally this displacement would be experimentally determined with a calibration experiment, but at the very least the possibility that it is not zero should be acknowledged. It would also be interesting to report on the measured values of the xy plane displacement.

We thank the reviewer for raising this point. To clarify our original statement, we intended to say that we expected the axial displacement, d_z , to be small with respect to the axial waist of the PSF (w_z equal to roughly 0.75 microns), such that $\exp(-(d_z/w_z)^2)$ would result in only a small correction to the ccRICS amplitude. However, the reviewer is correct that this displacement would ideally be experimentally determined. To address this, we have imaged (z-stacks) of immobilized fluorescent beads for the determination of the axial displacement and for measuring the xy plane displacements. We found that the factor $\exp(-(d_z/w_z)^2)$ is 1.0178 indicating that our initial assumption of this factor being small and thereby having negligible effect on our results is true. The details of the experiments and analysis are added in the methods (Section: Experimental determination of axial displacement, line 298) and the results in the SI (Figure S5).

Additional minor comments.

5) The model proposed by the authors seem to assume that the studied TFs passively bind to either "passive" or "active" DNA regions. But is it possible that they themselves participate in the activation of the DNA?

Thank you for giving us a chance to clarify this. Our results indicate the dependence of the pioneer factors (particularly Zld) on the chromatin structure for binding the DNA. Once bound, the pioneer factors can maintain the state of accessibility, a role that has been proposed previously¹. This leads to binding of other TFs that can regulate gene expression and "activate" the region. Although our results cannot accurately predict whether pioneer factor binding can lead to activation, this seems unlikely according to previous studies. For example: in *zld* mutants, target gene expression is delayed (until the concentration of the patterning factor, such as Dorsal, reaches a high value²)³. In contrast, no expression is observed if the patterning factor itself is removed. Also, the binding of the patterning factor (Dorsal) is reduced due to the removal of Zld, but Zld binding is not affected due to the removal of Dorsal¹. All these support a model in which the pioneer factors bind the DNA

and maintain a state of accessibility, facilitating the binding of other TFs, which can finally lead to transcriptional activation. This is also in agreement with a recent study that proposes that pioneer factors make the enhancers accessible, and this is followed by activation driven by correct combination of TFs. We have added an explanation for this in the concluding paragraph. See line 223.

6) The fit of the 2D ACFs should return not only the value of the TFs immobile fraction, but also the diffusivity of the mobile fraction. Why not report this value, and use it to estimate the limit value for k_{on} (in the "residence time of Zld" section)?

We thank the reviewer for the suggestion. We have now reported the value of diffusivity and found the value of diffusion limited k_{on} . Explanation added in the "Residence time of Zld" section (line 433) and explanation modified accordingly in line 150.

Reviewer 2:

SUMMARY OF THE ADVANCE MADE IN THIS PAPER AND ITS POTENTIAL SIGNIFICANCE TO THE FIELD

Dima and Reeves estimate the bound and unbound concentration of two critical early transcription factors in the *Drosophila* embryo during zygotic genome activation. The paper employs raster image correlation spectroscopy (RICS) to examine how Zelda-sfGFP and GAF-sfGFP change during zygotic genome activation (ZGA). They find that Zelda concentrations increase over the pre-ZGA cycles, but the amount bound to DNA is more constant. Conversely, they find that GAF binding increases dramatically in NC14. This work represents an orthogonal approach to previous work documenting both Zelda and GAF nuclear behavior through other forms of quantitative imaging and sequencing based approaches. It adds incrementally to our understanding of how these TFs change during ZGA.

SUGGESTIONS TO AUTHORS

Major:

How these results fit with previous literature should be more thoroughly discussed. ChIP-seq and an abundance of live imaging data already exist for both Zelda and to a lesser extent GAF. How do these findings compare to previous work? Especially about the known behavior of Zelda at transcriptional hubs and the changes in occupancy at specific sites from ChIP-seq data? The changes in total TF nuclear concentration are well documented, so the novelty of this work lies primarily in the measurements of the fraction bound.

We thank the reviewer for pointing this out. Although for most TFs (for example: the patterning factors) there are a plethora of reports on absolute concentrations and the dynamics based on live imaging, unfortunately this is not the case for the pioneer factors. We do mention this in the first paragraph of "Zld levels bound to DNA decrease while nuclear concentration increases" section (line 86), stating "Increase in Zld levels have been reported previously using immunoblotting." And while Zld ChIP-seq results lack the temporal resolution that our measurements have, they report a similar overall trend in binding. We have included this explanation in the fourth paragraph of the section: "Zld levels bound to DNA decrease while nuclear concentration increases" (line 125). Single molecule imaging study found a similar fraction of immobile Zld as we did, and we have discussed that in the second paragraph of the same section (line 98). Although the dynamics of Zld hubs are not well-documented due to the short timescale of the single-molecule experiments, single-molecule imaging and tracking videos⁴ show the rapid formation of hubs after mitosis, which is consistent with our finding, and we have added this in third paragraph of the same section (line 104).

GAF ChIP-seq also report a similar overall trend in binding (in very limited resolution) and we have included the explanation in the fourth paragraph (line 197) of the section "GAF levels increase suddenly in nc 14".

While in general the assumption that histones are in chromatin is well justified, in the early embryo there is a vast excess of maternally provided histone. Indeed, the majority of histone H3 is not bound to chromatin in NC11. The assumption that H2Av-RFP marks exclusively chromatin may not hold in the early embryo. The authors should consider how this might affect their interpretation of the ccRICS data.

We agree with the reviewer that there may be excess histone, especially in the earlier nuclear cycles. We had considered this possibility before our first submission of the manuscript and, through examining the equations we describe in the Methods section titled “Fitting ccRICS cross-correlation functions (CCFs) to estimate correlated binding,” we determined that unbound H2Av-RFP does not affect our ccRICS data.

To obtain the fraction of the sfGFP species correlated to H2Av-RFP, the amplitude of the CCF is divided by the amplitude of the ACF in the red channel. This effectively cancels out the total concentration of H2Av-RFP from the calculation. In equation (7) of the Methods, we denote the concentration of H2Av-RFP not correlated to the sfGFP species as \bar{c}_3 . The equations are agnostic as to whether the species in the pool denoted by \bar{c}_3 is bound to DNA, but not “near” an sfGFP species, or whether it is not bound to DNA at all. To clarify this, in the Methods (lines 373), we now include a sentence describing the possible compositions of the pool denoted by \bar{c}_3 .

Minor:

In the abstract and results it would be helpful to be more precise about what period is under consideration (NC10-NC14) when discussing the TF concentration changes. It is clear in the figures, but not specified in the text until very late.

Thanks for pointing this out. We specified the time period in the abstract (line 29) and in the beginning of the “Zld levels bound to DNA decrease while nuclear concentration increases” (line 85).

Given the importance of line scan time to the RICS method it should be reported (not just frame time) in the methods.

The line time was 5 ms and can be calculated from the frame time by dividing by the number of rows in the image (1024). We have added this explanation in the Methods (“Live imaging” section, line 276).

There is a typo: “This raises the question of whether other factors are present in during the major wave”.

Corrected.

References

1. Sun, Y. *et al.* Zelda overcomes the high intrinsic nucleosome barrier at enhancers during *Drosophila* zygotic genome activation. *Genome Research* **25**, 1703-1714 (2015).
2. Kanodia, J. S. *et al.* Pattern formation by graded and uniform signals in the early *Drosophila* embryo. *Biophysical Journal* **102**, 427-433 (2012).
3. Nien, C. Y. *et al.* Temporal coordination of gene networks by Zelda in the early *Drosophila* embryo. *PLoS Genetics* **7**, (2011).
4. Mir, M. *et al.* Dynamic multifactor hubs interact transiently with sites of active transcription in *Drosophila* embryos. *Elife* **7**, e40497-e40497 (2018).

Second decision letter

MS ID#: dev.204460R1

MS TITLE: Bulk-level maps of pioneer factor binding dynamics during *Drosophila* maternal-to-zygotic transition

AUTHORS: Sadia Siddika Dima and Gregory T. Reeves

Dear Dr Reeves,

I have now received all the referees reports on the above manuscript, and have reached a decision. The referees' comments are appended below, or you can access them online: please go to .

The overall evaluation is positive and we would like to publish a revised manuscript in Development, provided that the referees' comments can be satisfactorily addressed. Please attend to all of the reviewers' comments in your revised manuscript and detail them in your point-by-point response. If you do not agree with any of their criticisms or suggestions explain clearly why this is so. If it would be helpful, you are welcome to contact us to discuss your revision in greater detail. Please send us a point-by-point response indicating your plans for addressing the referees' comments, and we will look over this and provide further guidance.

Reviewer 1

Advance summary and potential significance to field

Providing dose-response curves for the two important pioneer factors, and showing those are qualitatively different, represents a meaningful contribution to the field.

Comments for the author

Minor:

- Writing could be clarified in some places. For example, line 106: "It should be noted that the presence of hubs results in Zld particles with higher brightness than Zld monomer, given Zld is unlikely to form multimers [45,46,50,51]". I do not understand the logical connection between the two halves of this sentence.
- Lines 135 - 152: It is unclear from this paragraph if the measured KD corresponds to binding to active region, inactive region, or both. Although this is somewhat explained in the methods, it would be good to also clarify this when presenting the results. For example, why not use the symbols K_{Da} and K_{Di} introduced in the methods section? Related to this, it would also be good to explain in the figure caption what the lines are in Fig. 2E and 3E.
- Lines 145: I don't think the choice of using a coefficient of 2 in the Hill function used to fit binding curves is justified anywhere. Why a Hill function in the first place, and why a coefficient of 2? Moreover, the model used to fit the binding curves (Eq. 10, line 386) seems in fact to just be a non-cooperative model (Hill coefficient of 1). So which equation exactly was used to fit the data in Figs. 2E and 3E?
- Lines 426-436: Thank you for adding Zld's diffusion coefficient and diffusion-limited rate. However this section now seems to have contradictory statements, where it is first stated that the diffusion limited on-rate is on the order of 10^{-4} /nM/s, and then (after having calculated it) that it is on the order of $10^{\dagger-3}$. Where does the first estimate come from, and is it valid in this case?
- Line 126: Is ChIP-Seq suppose to detect only binding corresponding to active binding?
- line 125: "of" not "if"
- Some of the references are missing page numbers (e.g. 57, 60, etc..)

Second revision

Author response to reviewers' comments

Comments from the Reviewers:

Reviewer 1: SUMMARY OF THE ADVANCE MADE IN THIS PAPER AND ITS POTENTIAL SIGNIFICANCE TO THE FIELD

Providing dose-response curves for the two important pioneer factors, and showing those are qualitatively different, represents a meaningful contribution to the field.

SUGGESTIONS TO AUTHORS

Minor:

- Writing could be clarified in some places. For example, line 106: "It should be noted that the presence of hubs results in Zld particles with higher brightness than Zld monomer, given Zld is unlikely to form multimers [45,46,50,51]". I do not understand the logical connection between the two halves of this sentence.

We appreciate the suggestion to clarify this. We meant that populations with increased brightness are due to the presence of hubs (as reported earlier (1, 2)) since other populations such as dimers that could cause such increased brightness are unlikely to form in case of Zld. As this may have caused confusion for most readers, and is not an important explanation for what we did in the paper, we have modified the sentence in line 120 to remove that information.

- Lines 135 - 152: It is unclear from this paragraph if the measured KD corresponds to binding to active region, inactive region, or both. Although this is somewhat explained in the methods, it would be good to also clarify this when presenting the results. For example, why not use the symbols KDa and KDi introduced in the methods section? Related to this, it would also be good to explain in the figure caption what the lines are in Fig. 2E and 3E.

We agree with the reviewer that the suggested symbols will be helpful for the readers. In the mentioned lines the KD corresponds to the active region. We added explanation to lines 153, 162. Changed the symbols to KDa and KDi for both Zld and GAF (lines 154, 163, 226, 227, and 452). We added explanation for the lines in the captions of Fig. 2E and 3E as well.

- Lines 145: I don't think the choice of using a coefficient of 2 in the Hill function used to fit binding curves is justified anywhere. Why a Hill function in the first place, and why a coefficient of 2? Moreover, the model used to fit the binding curves (Eq. 10, line 386) seems in fact to just be a non-cooperative model (Hill coefficient of 1). So which equation exactly was used to fit the data in Figs. 2E and 3E?

We thank the reviewer for raising this point. Hill function is widely employed in mathematical models of gene expression studies and it explains our data well. Hill coefficient of 2 was used in the mentioned curve for inactive population of Zld due to the sigmoidal nature of the data points (see Figure 2E). In all the other binding curves (active Zld, both active and inactive GAF) the data points are hyperbolic, thus a Hill coefficient of 1 gives good fit (Figures 2E and 3E). We have added the Hill coefficient, n in Eq. 10, line 414 to make it more generalized and added the explanation in lines 161 and 417.

- Lines 426-436: Thank you for adding Zld's diffusion coefficient and diffusion-limited rate. However this section now seems to have contradictory statements, where it is first stated that the diffusion limited on-rate is on the order of 10^{-4} /nM/s, and then (after having calculated it) that it is on the order of 10^{-3} . Where does the first estimate come from, and is it valid in this case?

Sorry for the confusion. The rule of thumb is that $10^{-4} < k_{on} < 10^{-3} \text{ nM}^{-1}\text{s}^{-1}$ (3) is for protein-ligand binding. The k_{on} is much higher ($0.1 \text{ nM}^{-1}\text{s}^{-1}$) for 1D DNA sliding. This large variation in the rule of thumb indicates that an alternative relation, which is more appropriate for protein-DNA binding, needs to be considered. So, we used the Smoluchowski equation, which gave us a $k_{on} \sim 10^{-3} \text{ nM}^{-1}\text{s}^{-1}$ (resembling the upper limit of the rule of thumb). We have modified the explanation in the "Residence time of Zld" section.

- Line 126: Is ChIP-Seq suppose to detect only binding corresponding to active binding?

We thank the reviewer for giving us a chance to clarify this, and we agree with the reviewer that ChIP mainly represents the active regions. Ideally, ChIP-Seq should detect all the Zld populations bound to DNA. However, tightly packed inactive regions are sheared to a lesser extent during the DNA fragmentation step, thereby resulting in lower detection levels of these regions in ChIP-Seq. To detect these regions, longer fragments (350~800 bp) need to be re-fragmented (4). However, for

Zld ChIP-Seq, 100 to 300 bp fragments were used. Thereby, detected peaks correspond primarily to active regions (see Figure 1D in (5)). All together, we expect the ChIP to be heavily enriched in the active regions. We added brief explanation in line 138.

- line 125: "of" not "if"
Corrected.

- Some of the references are missing page numbers (e.g. 57, 60, etc.).
We thank the reviewer for pointing this out. We have revised and corrected all the references. Additionally, we have also changed the reference style to meet the journal's requirement.

References

1. M. Mir, A. Reimer, J. E. Haines, X.-Y. Li, M. Stadler, H. Garcia, M. B. Eisen, X. Darzacq, Dense Bicoid hubs accentuate binding along the morphogen gradient. *Genes & development* **31**, 1784-1794 (2017).
2. M. Mir, M. R. Stadler, S. A. Ortiz, C. E. Hannon, M. M. Harrison, X. Darzacq, M. B. Eisen, Dynamic multifactor hubs interact transiently with sites of active transcription in *Drosophila* embryos. *Elife* **7**, e40497-e40497 (2018).
3. K. D. Wittrup, B. Tidor, B. J. Hackel, C. A. Sarkar, *Quantitative Fundamentals of Molecular and Cellular Bioengineering* (The MIT Press, Cambridge, MA, 2019).
4. R. Nakato, K. Shirahige, Recent advances in ChIP-seq analysis: from quality management to whole-genome annotation. *Brief Bioinform*, bbw023 (2016).
5. M. M. Harrison, X. Y. Li, T. Kaplan, M. R. Botchan, M. B. Eisen, Zelda binding in the early *Drosophila melanogaster* embryo marks regions subsequently activated at the maternal-to-zygotic transition. *PLoS Genetics* **7** (2011).

Third decision letter

MS ID#: dev.204460R2

MS TITLE: Bulk-level maps of pioneer factor binding dynamics during *Drosophila* maternal-to-zygotic transition

AUTHORS: Sadia Siddika Dima and Gregory T. Reeves

Dear Dr Reeves,

I am happy to tell you that your manuscript has been accepted for publication in *Development*, pending our standard publication integrity checks.